# Global motion processing in infants' visual cortex and the emergence of autism

Irzam Hardiansyah [1✉], Pär Nyström[2], Mark J. Taylor[3], Sven Bölte [1,4], Angelica Ronald [5] & Terje Falck-Ytter [1,6,7✉]

Autism is a heritable and common neurodevelopmental condition, with behavioural symptoms typically emerging around age 2 to 3 years. Differences in basic perceptual processes have been documented in autistic children and adults. Specifically, data from many experiments suggest links between autism and alterations in global visual motion processing (i.e., when individual motion information is integrated to perceive an overall coherent pattern). Yet, no study has investigated whether a distinctive organization of global motion processing precede the emergence of autistic symptoms in early childhood. Here, using a validated infant electroencephalography (EEG) experimental paradigm, we first establish the normative activation profiles for global form, global motion, local form, and local motion in the visual cortex based on data from two samples of 5-month-old infants (total $n = 473$). Further, in a sample of 5-month-olds at elevated likelihood of autism ($n = 52$), we show that a different topographical organization of global motion processing is associated with autistic symptoms in toddlerhood. These findings advance the understanding of neural organization of infants' basic visual processing, and its role in the development of autism.

[1] Center of Neurodevelopmental Disorders at Karolinska Institutet (KIND), Department of Women's and Children's Health, Karolinska Institutet & Stockholm Health Care Services, Stockholm, Sweden. [2] Uppsala Child and Baby Lab, Department of Psychology, Uppsala University, Uppsala, Sweden. [3] Department of Medical Epidemiology & Biostatistics, Karolinska Institutet, Stockholm, Sweden. [4] Curtin Autism Research Group, Curtin School of Allied Health, Curtin University, Perth, Australia. [5] Department of Psychological Sciences, Centre for Brain and Cognitive Development, Birkbeck, University of London, London, UK. [6] Development and Neurodiversity Lab, Department of Psychology, Uppsala University, Uppsala, Sweden. [7] Swedish Collegium for Advanced Study, Uppsala, Sweden. ✉email: irzam.hardiansyah@ki.se; terje.falck-ytter@psyk.uu.se

Imagine looking at a flock of birds forming a cyclic motion in the sky. Now imagine how this sight would be if you could not perceptually combine the motions from the individual birds. Global motion refers to the perceived direction of a dynamic input when that direction is the result of a combination of many individual motion signals within the stimulus[1]. This process recruits extrastriatal areas including the middle temporal MT (V5) area located in the lateral occipital lobe[2–4]. Analogously, global form refers to perceived coherent structure from static local visual signals, which is subserved by visual areas along the ventral stream (e.g., V4, intraparietal sulcus/IPS)[4,5]. People with Autism Spectrum Disorder (ASD; hereafter autism) tend to perform differently on global motion tasks relative to individuals without autism (reviewed in ref.[6]). In autistic people, perception of global motion has been associated with social perception (e.g., biological motion and facial emotion processing) in multiple studies[6–10]. It is conceivable that atypicalities in global motion processing early in life are part of developmental cascades leading to autism in toddlerhood. Yet, no study has tested whether differences in global motion processing precede the onset of autistic symptoms.

In typical development, global motion processing emerges earlier than global form processing but takes extended time to fully develop. Around the age of 7 weeks, infants begin to show sensitivity to visual motion signals and very soon afterwards, around 12 weeks of age, their ability to integrate these local signals into coherent percept can be clearly established by both behavioral and electrophysiological measures[11]. However, it is not until they reach adolescence that this perceptual ability attains adult-like levels[11,12]. In contrast, global form processing is hardly detectable at the age of 4 months, yet it attains maturity much earlier and is thought to be less vulnerable to developmental perturbations[11,12].

This study combined cohorts of 5-month-old infants from two different studies EASE ($N_{EASE}$ = 73) and BATSS ($N_{BATSS}$ = 452; see Methods for further information). The EASE sample, which included a group at elevated likelihood of autism (due to familial history of the condition; EL, $n_{EL}$ = 52) and a low likelihood group (no familial history of autism; LL, $n_{LL}$ = 21) was used to probe relations between visual processing and later autism symptoms (measured via clinical observation in toddlerhood). All infants in the EL group had an older sibling with a confirmed ASD

diagnosis[13]. Both the BATSS sample and the LL group from the EASE sample consisted of infants without increased likelihood of autism, and together these groups comprised 473 individuals. This combined group is not representative of the entire population in all respects[14], but given its size and the basic nature of the phenotype in question (basic visual processing in infancy), we termed it Normative Sample. In addition to contributing to such normative reference data, the BATSS sample, consisting of monozygotic (MZ) and same sex dizygotic (DZ) infant twins, was used to study the contribution of genetic and environmental influences to the variation in the included EEG phenotypes. Finally, because autism is thought to represent the extreme end of a continuum, the BATSS sample also provided an opportunity to test whether any link between EEG phenotypes and later symptoms in the EASE sample, would replicate at a trait level in the BATSS sample. To accomplish this, we measured individual differences in social communication and autistic traits via parent reports in the BATSS sample in their second year of life. Thus, including the EASE and the BATSS sample in the same study provided several analytic and conceptual opportunities beyond what is typically done in studies of early signs of autism.

All infants participated in a previously validated infant EEG experiment[15,16] employing a combination of visual stimuli that formed four experimental conditions: Global Form, Global Motion, Local Form, and Local Motion (Fig. 1a; Methods). The analyses below were based on the visual evoked responses in the visual cortex under these four conditions.

We recently discovered that infants at low likelihood for autism had more centralized EEG activation to global visual information compared to infants at elevated likelihood for autism, whose activation tended to be more lateralized[15]. Therefore, in the current paper, which includes the same infants but with outcome data as well, we related the lateralized activation patterns across electrodes covering the visual cortex to core autism symptoms, as measured by the Autism Diagnostic Observation Schedule 2nd edition (ADOS-2) Comparison Score (CS);[17] see also Methods. The ADOS-2 is a gold standard diagnostic instrument for ASD based on direct observation of the child during social interaction, led by experienced child psychologists. To study conceptually similar associations in the BATSS sample of normative infant twins, we used the Infant Toddler Checklist[18] (ITC; measures social communication) and the Quantitative Checklist for Autism

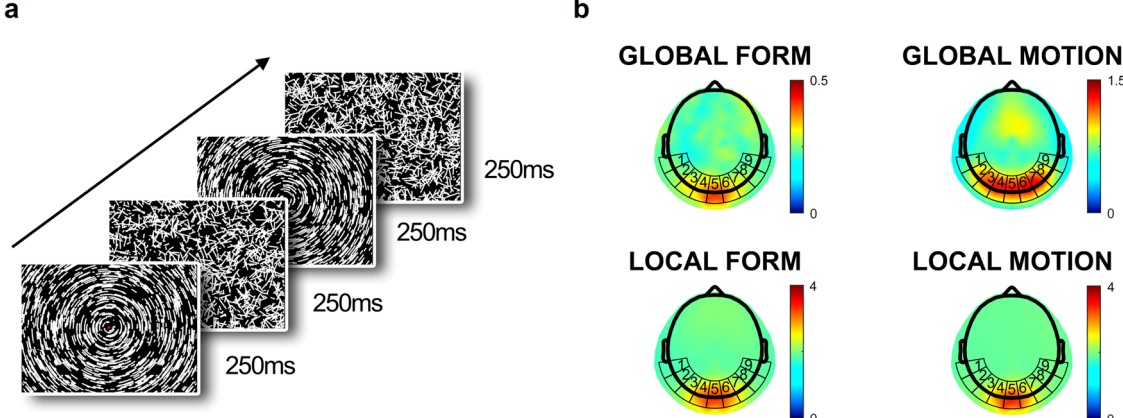

**Fig. 1 Experimental stimuli and scalp topographies from typically developing infants. a** Visual stimuli showed interchanging coherently and randomly oriented line segments (form conditions; shown here) or coherently and randomly moving dots (motion conditions; see Supplementary Video 1) to evoke global and local processing, respectively. **b** Overhead mean topographical maps for the four conditions based on the Normative Sample of 5-month-old infants. Distinctive visual cortex activation topographies were observed in both Global Motion and Global Form conditions (statistics in main text); however, visibly widespread activity across bilateral lateral electrodes were observed only in the former. Local Form and Motion produced largely similar activation concentrated on the medial electrodes.

in Toddlers[19] (QCHAT; see Methods). As in the previous report[15], the main variable of interest—termed the Laterality Score – represents the differential activity of the lateral relative to the central electrodes covering the visual cortex.

## Results

**Scalp topographies in response to basic visual stimuli in normative infants.** We first studied how global form and motion, as well as local form and motion, activates the cortex in typical development, using data from the Normative Sample. Fig 1b shows topographical activation of visual cortex, as represented by the T2circ statistic[15,20], in response to each of the four stimulus conditions plotted across nine Areas of Interest (AOIs) covering the approximate area of the visual cortex (see Methods). The Global Motion condition produced a more dispersed topography, as opposed to the more centralized topographies in the other three conditions. Planned comparisons of the four conditions using the L2-distance scores of topography-vectors (see Methods) confirmed that the two local change (Local Form and Motion) conditions were similar ($p > 0.25$) while the two global change conditions were each distinctive (Global Form: ***$p < .001$ vs. all other three conditions; Global Motion: ***$p < 0.001$ vs. all other three conditions). See Supplementary Table 1 and Supplementary Fig. 1.

**Association between lateralized activation and subsequent autistic symptoms in the EASE sample.** In the EASE sample (see Methods and Supplementary Tables 2–4), a significant positive main effect of Laterality Scores of Global Motion at 5 months was found on ADOS-2 total scores at 36 months (Fig. 2a, Fig. 3a, Supplementary Table 1). These results were replicated for ADOS-2 total scores measured earlier at 24 months, among all infants who attended both ADOS-2 sessions (at 24 and 36 months) and, separately, among those who only attended this session alone (not at 36 months); see Fig. 2a (inset), Methods, and Supplementary Table 1. It is notable that the variability in ADOS scores is considerable also in the LL group, this is however quite typical and reflects in part the fact that the ADOS is not originally designed to study trait variation in non-referred samples.

To investigate clinical relevance, we split the EL group into two subgroups using previously established cut-offs for the ADOS-2[21,22]. Planned group comparisons of Global Motion Laterality Scores showed that the EL high-ADOS group (with ADOS-2 CS scores ≥5 corresponding to moderate to high clinical concern[21,22]) was different from all other groups; see Fig. 2b, Supplementary Table 1, and Methods. No group difference was found in the other three stimulus conditions (Fig. 3b–d, Supplementary Table 1). These results also confirmed the similarity of the larger twin sample to the EASE LL control group, and hence the use of these groups in the Normative Sample.

Next, we tested differences at specific AOI locations between the high-ADOS group and the Normative Sample for each condition (see Methods). Only in Global Motion condition the two groups differed; the EL high-ADOS group showed stronger activation in the left lateral electrodes and lower activation centrally; see Fig. 2c (see also Supplementary Fig. 2). Details of the statistical tests and contrasts can be found in Supplementary Table 1.

**Twin modeling.** To investigate genetic and environmental influences on individual differences in the visual cortical EEG lateralization, we analyzed the twin data within a standard behavioral genetics framework, comparing similarity within MZ and DZ twins (see Methods and Supplementary Tables 2-4).

Univariate twin modeling suggested a moderate broad-sense heritability ($h^2 = 0.226$, 95%-CI: 0.047–0.392, $n = 366$) of the Laterality Scores in the Global Motion condition. Individual differences in the other three conditions were found to be influenced by a unique environmental factor alone, which includes measurement error. This indicates that while these three conditions are associated with clear group level activation patterns, individual differences in the Laterality Scores may not be reliable. Raw observable correlations and model output are provided in Supplementary Table 1.

**Association with autistic traits in the BATSS sample.** In the BATSS sample we used the four Laterality Scores as regressors, and the ITC total score at 14 months and the QCHAT score at 24 months as the outcome in two separate GEE models (see Methods and Supplementary Tables 2–4). The four Laterality Scores together explained only a very small amount of variance (<2%) in either of the outcome variables, suggesting that the results of these analyses may not be meaningful to interpret (all details in Supplementary Table 1).

## Discussion

What role perceptual differences play in autism has been discussed for decades, but empirical studies focused on older children and adults[6,23,24]. The present study suggests that alterations in the neural organization of basic perceptual processes are present in young infants who later show symptoms of autism. More specifically, the results indicate that such alterations are linked to processing of visual global motion, and not to processing of global form or local visual change, indicating its specific involvement in the developmental cascades toward autism.

Besides its implications for autism, the study provides data from 473 infants expected to develop typically, on activation patterns in the visual cortex during visual form and motion processing. Although one needs to be cautious in interpreting the source of the signals detected by EEG, the results imply greater activation of the lateral areas of visual cortex in global motion processing, compared with the other three conditions (see e.g., section "F1 as a Signature of Global Processing" in Supplementary Information of ref. [16] for a detailed discussion about this issue linked to our experimental paradigm). This, in turn, may support the hypothesis that the extrastriate area MT (V5) in the visual cortex supports processing of globally coherent motion in human infants[12,16]. Both V1 and MT (V5) have been implicated in multiple studies of visual processing in autism previously[25–28].

There are at least three (not necessarily mutually exclusive) potential explanations for the different pattern seen in EL high-ADOS group. First, a global motion processing activation pattern highly similar to what observed for EL high-ADOS group has been reported in typical adults during scotopic (low lights) viewing conditions, using the same stimuli as the current study[29]. While this could implicate overreliance on magnocellular processing[30], it is notable that we did not observe a parallel reduction in amplitude in the form condition[29], rendering this explanation unlikely (Fig. 4a and Supplementary Table 1 for a supporting analysis). Second, again using the same experimental paradigm as in the current study, Wattam-Bell and colleagues[16] found largely similar visual cortical topographical patterns during both global motion and global form processing in typically-developing infants as what observed in our Normative Sample. They attributed the more lateralized activation during global motion in infants compared to adults to reduced inhibition between striatal and extrastriate areas, due to cortical immaturity[31]. Following this reasoning, the global motion pattern seen in the EL high-ADOS group could suggest even less

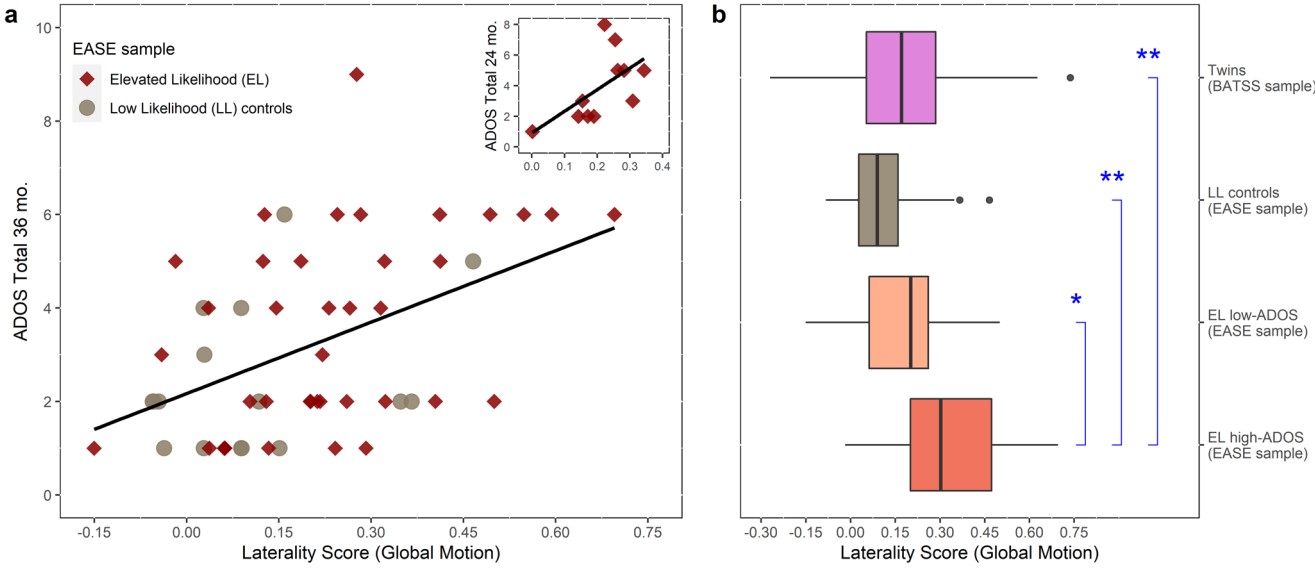

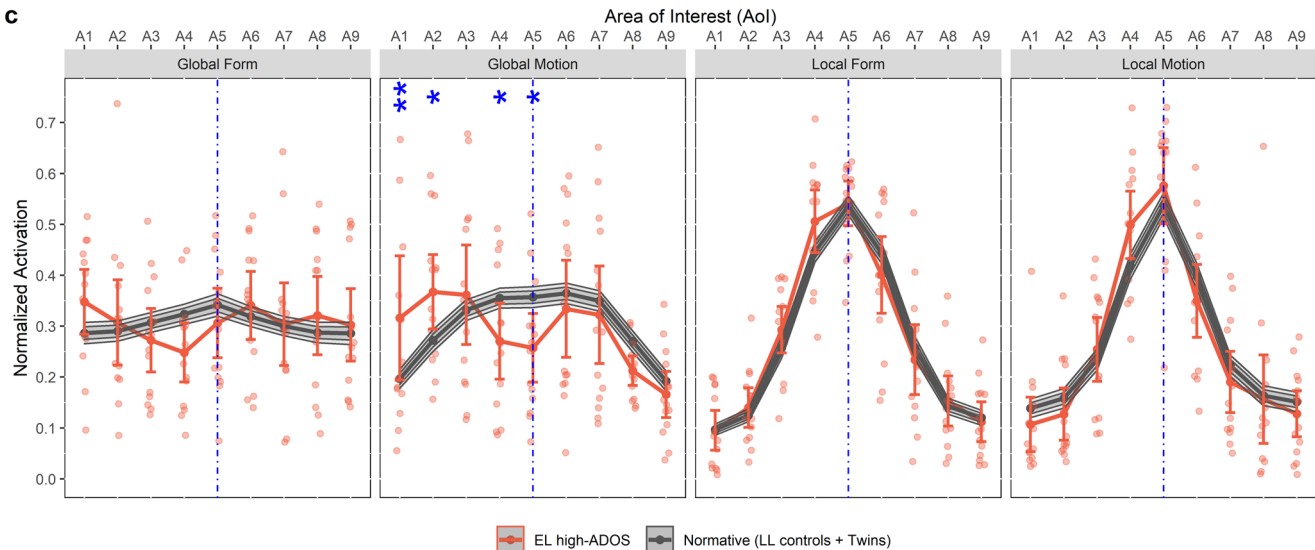

**Fig. 2 Topographical organization of global motion processing is different in infants who have high levels of autism symptoms in toddlerhood. a** More lateralized activation in the coherent motion condition was associated with higher autistic symptoms (ADOS-2 total scores at 36 months) in the overall EASE sample ($r = 0.465$, ***$p < 0.001$, $n = 55$), as well as in the EASE EL group ($r = 1.448$, **$p = 0.004$, $n = 39$). These were confirmed with a linear regression model ($\beta = 7.22$, **$p = 0.005$, partial $\eta^2 = 0.170$, $n = 55$) which also revealed no moderation by sex or autism-likelihood group (both $p > 0.20$). No association was found for the other conditions (all $p > 0.25$). **a** inset. Replication in an independent sub-sample of infants at elevated likelihood of autism who partook in ADOS-2 session at 24 months but not at 36 months ($r = 0.596$, *$p = 0.027$, $n = 11$, one-tailed). **b** Infants who went on to develop high levels of autistic symptoms at 36 months (EL high-ADOS, $n = 14$) according to pre-established clinical cut offs (see main text) had more lateralized global motion processing than infants at elevated likelihood with few autistic symptoms (EL low-ADOS: $d = 0.25$, *$p = 0.029$, $n = 25$), infants at low likelihood for autism (LL controls: $d = 0.33$, **$p = 0.002$, $n = 16$) and the larger community twin sample (BATSS: $d = 0.31$, **$p = 0.003$, $n = 452$), but no difference was found between these three latter groups (all $p > 0.50$). Boxplots show the sample median, and the first and third quartiles; whiskers show minimum and maximum (±1.5 s.d.); dots are outliers (±1.96 s.d.). Planned group comparisons done with linear mixed model and EMM effect-size analysis. **c** Comparison of activation across the nine AOIs between the Normative Sample and EL-high ADOS group. During global motion processing, EEG activity in the latter group was characterized by atypically strong extreme left lateral activation (AOIs A1 and A2: $M = -0.120$, **$p = 0.009$, $d = -0.10$ and $M = -0.095$, *$p = 0.026$, $d = -0.08$, respectively) and atypically weak midline activation (AOIs A4 and A5: $M = 0.085$, *$p = 0.044$, $d = 0.07$ and $M = 0.100$, *$p = 0.026$, $d = 0.08$, respectively). (Notes: *$p < 0.05$; **$p < 0.01$; ***$p < 0.001$, for blue asterisks in **b** and **c**; all $p$ values are Benjamini-Hochberg-corrected). Data from the Normative Sample group ($n = 473$) is plotted as mean, 95%-C.I. (inner band, darker shade) and 99.5%-C.I. (outer band, lighter shade). Data from the EL high-ADOS group ($n = 14$) are plotted as mean & 95%-C.I. (error bars) along with the individual data points (shaded). More detailed activation plots for the Normative Sample are provided in Supplementary Fig. 2. Observe here that confidence intervals for the Normative Sample are comparatively very small due to its much bigger sample size.

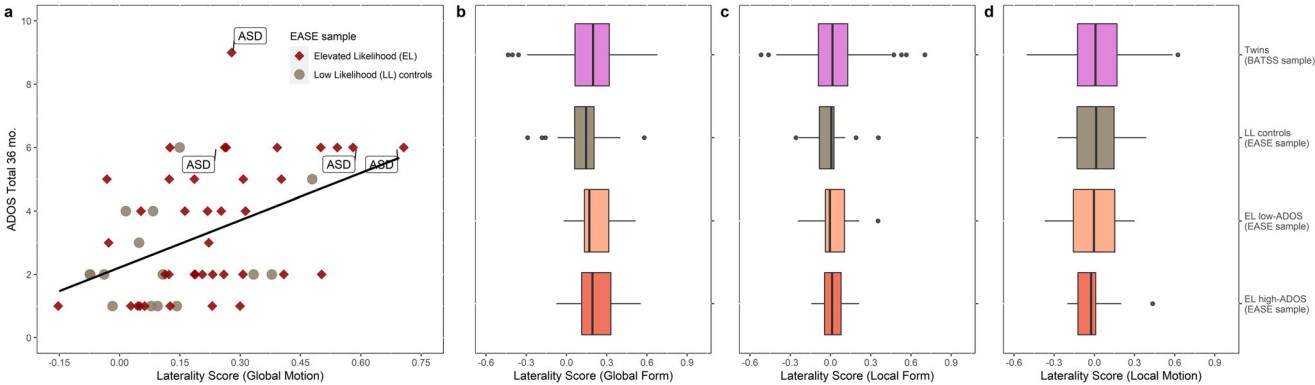

**Fig. 3 ASD diagnoses among EASE participants and group differences of Laterality Scores in the three other stimulus conditions. a** Laterality Scores in the Global Motion condition versus ADOS-2 CS total scores at 36 months from EASE children who partook in the DSM-5 (Diagnostic and Statistical Manual of Mental Disorders 5ed) diagnostic session ($n = 55$), highlighting very few positive cases ($n = 4$) with a confirmed ASD diagnosis. **b** No difference in the mean Laterality Scores between the four groups of infants during global form processing (all $p > 0.25$). **c**. and **d**. Similarly, mean Laterality Scores between the four infant groups were found to be very similar during both the local form processing (all $p > 0.50$) and local motion processing (all $p > 0.50$), respectively. Boxplots show the sample median, and the first and third quartiles; whiskers show minimum and maximum ($\pm 1.5$ s.d.); dots are outliers ($\pm 1.96$ s.d.); sample sizes are the same as reported in Fig. 2b ($n_{EL\ high-ADOS} = 14$, $n_{EL\ low-ADOS} = 25$, $n_{LL} = 16$, $n_{BATSS} = 452$). Complete statistics in Supplementary Table 1.

inhibition between the two cortical areas, perhaps due to altered GABAergic signaling in autism[23,32]. However, it is worth noting that we observed no effect of age on the Laterality Score for Global Motion, in either EASE or BATSS infants, (see Supplementary Table 1 for a supporting analysis) despite our large infant sample of ages from 131 to 207 days. Third, primate studies have shown that MT (V5) is one of the earliest developing visual areas of the brain, driven by direct thalamic (from the pulvinar and geniculate nuclei) input bypassing the V1 in the initial stage[2,33,34]. In typical development, projections from V1 supersede the earlier subcortical connections in postnatal life and eventually dominate the input[2,33]. Our findings suggesting distinct organization of the visual cortex during global motion processing in the EL high ADOS group could implicate that the transition from subcortico-cortical to cortico-cortical processing is delayed or different in this population[35] (a similar idea has been proposed earlier for face processing in autism[36,37]).

At a group level, atypical activation was more pronounced in the left hemisphere than the right, which is broadly in line with previous findings of atypical brain asymmetries in autism[38]. As shown in (Fig. 4b), at individual level, EEG activation patterns for most infants in the EL high-ADOS group suggested an elevated extrastriatal to striatal activation predominantly on one side (not both). Importantly, our results indicate that the topographical organization of local visual processing is unperturbed in autism (Fig. 2c), a finding which is in line with studies with older age groups (i.e., adolescents and adults)[39–41].

What could be the consequences of different global motion processing in infancy? Difficulties in processing global motion have been linked to problems with perception of biological motion[6,8–10] and emotional expressions[7,10] in autism, both of which are linked to the superior temporal sulcus which receives dense projections from MT[42,43]. Thus, it is possible that early differences in global motion processing affects these phenotypes in the social domain. Further, efficient motion processing is critical for emerging manual action abilities (e.g., reaching/grasping)[43,44]. Thus, our finding provides a potential explanation for both basic perceptual-action and socio-cognitive differences in autism—hypotheses that can be followed up in future studies.

Current views of neurobiology of autism posit widespread and multi-level brain developmental issues such as hyper-expansive regional cortex growth[45–47], altered brain connectivity[48,49], and

excitation/inhibition imbalance[50,51]. Both the lack of visual cortical inhibition and altered subcortical influence explanations proposed above could fit with these views. Nonetheless, it is also notable that none of the general theories predicts specific differences in global motion processing. We should emphasize that the atypical brain activation during global motion processing observed here does not necessarily indicate deficits in perceptual processing, although previous research may support such an interpretation[6,10,11,52]. It is noteworthy that a recent study of transcriptomic dysregulation in autism found most pronounced effects in the visual cortex[53].

It is worth noting that our stimuli did not include noise (additional masking dots), but rather showed alternating 100% coherent motion and 100% random motion. Thus, our results cannot be explained by overfitting to noise contained within the stimuli themselves[54,55]. Another stimulus property worth considering is the pace of presentation (250 msec for each alternating segment). It is conceivable that the result would have been less pronounced if we gave the infants more time to process the stimuli (see refs. [23,52] for data suggesting that performance is less different for longer presentation times).

Autism is generally conceived as the extreme end of a continuum of heritable traits encompassing the whole population[56]. Indeed, we also found the Laterality Score for global motion processing to be modestly heritable in the typical infant twin population represented by the BATSS sample. Nonetheless, in BATSS, we failed to find meaningful association between the Laterality Scores in any condition with either social communication development at 14 months (ITC Total Scores) or autistic traits at 24 months (Q-CHAT; all statistics in Supplementary Table 1). We offer two explanations, that are not mutually exclusive, for these results: a) it is possible that the association between global motion processing and later autism is more pronounced in infants at elevated likelihood due to more variability in early stage processing and later neurodevelopmental trajectories, and b) specifically for QCHAT, parent ratings of autistic traits (in BATSS) are less reliable[57] than clinician rated symptoms based on direct observation (in EASE).

This study is not without limitations. First, due to the young age of the infants, we do not have a direct measure of global motion perception to confirm whether or not the atypical pattern seen in the EEG reflects experiential or behavioral differences.

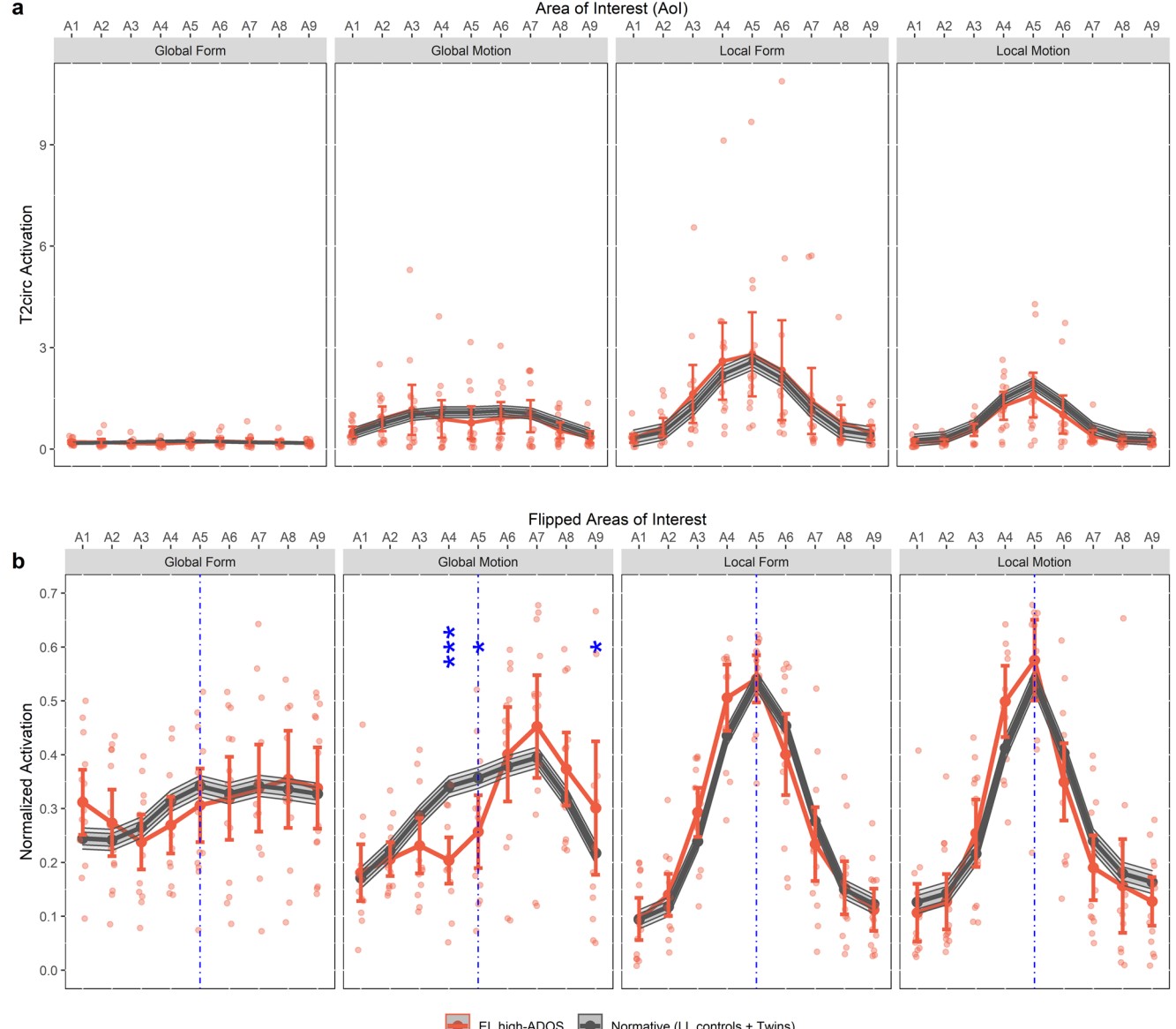

**Fig. 4 Nominal (unnormalized) and unilaterality of visual cortex activation during the four stimulus conditions. a** The non-normalized version of plots in Fig. 2c, showing the actual $T^2_{circ}$ activation values for all the four stimulus conditions. No quantifiable difference was evident between the Normative Sample and EL high-ADOS group in any of these conditions (all $p > 0.50$). Furthermore, the strength of activation during global form processing was so much weaker compared the other three conditions, suggesting the relative immaturity of this perceptual function at 5 months of age, in line with previous findings (e.g., refs. [16,17,29]) in the literature. **b**. The same plots (Fig. 2c) with the order of the nine AOIs reversed for some of the infants such that all the peak activations now lie on the same side (the right side, arbitrarily chosen). The term flipped AOIs in the figure refers to this rearrangement for some infants. As in the main result (Fig. 2c), midline activations (AOIs A4 and A5) were significantly weaker in EL high-ADOS (***$p < 0.001$, $d = 0.12$, and *$p = 0.018$, $d = 0.09$, respectively), while the extreme lateral activation (AOI A9) was stronger in EL high-ADOS (*$p = 0.048$, $d = -0.07$). Data from the Normative Sample group ($n = 473$) is plotted as mean, 95%-C.I. (inner band, darker shade) and 99.5%-C.I. (outer band, lighter shade). Data from the EL high-ADOS group ($n = 14$) is plotted as mean and 95%-C.I. (error bars) along with the individual data points. Confidence intervals for the Normative Sample group are comparatively very small due to its much bigger sample size. See also Supplementary Fig. 2 for more detailed AOI activation plots of the Normative Sample group and Supplementary Table 1 for complete statistics.

However, it has been demonstrated in earlier studies using the same type of stimulus as in the current report, that the activation of the visual cortex tracks the amount of coherence in the stimulus, in both adults and infants[58,59]. Specifically, because the amplitude of the 2 Hz-rhythm in the spectrum of SSVEP induced by the stimuli varies linearly with coherence level of the stimuli and is absent at 0% coherence (i.e., when all the dots are moving in random directions or all line segments arranged in random orientations)[58–60], this component is dependent on the presence of a coherent structure and, by the design of the stimuli, will only

be generated if the infant is sensitive to the transition from random to coherent configurations that happen twice every second. Second, we did not have sufficient sample size in the EASE sample to study diagnostic outcomes (but see Fig. 3a). Thus, further study is needed to replicate the current findings and link processing of global motion to later diagnostic outcomes. While the ADOS is a crucial part of diagnostic assessments for autism, a clinical diagnosis builds on more information (e.g., interview with parents). The group comparison in Fig. 2c, d reflects clinical concern based on the ADOS score, but as can be seen in Fig. 3a,

only some of the children who scored above the cut off met all criteria for a formal clinical DSM-5 diagnosis. Third, while labeled the Normative Sample in our study, it is indeed likely, given its large size, that some of the children included in the BATSS sample will develop developmental conditions, including autism. Still, it is probable that the results from this group represents variability corresponding approximately to that seen in the whole infant population in Sweden. Fourth, the results for global form are somewhat difficult to interpret due to the much weaker signal in this condition compared to global motion (presumably reflecting the immature state of global form processing at five months in typical development[60], see unnormalized plots in Fig. 4a). Lastly, while the BATSS twin sample helped us assess the heritability of global motion processing, it is difficult to know whether the lack of heritability for the other conditions reflects a true null effect or less variability (more centralized activation patterns overall; Fig. 1b, Fig. 2c) in these conditions.

In sum, our results suggest that the early differences in global motion processing could represent an important factor in the development of autism. Although we do not know the degree of specificity in the neural processes underlying the current results, the findings are generally in line with a 'sensory first' view of autism[23], particularly considering the fundamental role global motion processing is thought to have for everyday visual perception of objects and events and dynamic action development[43,44]. The currently identified antecedent marker can be exploited in future animal work to address causal hypotheses (e.g., marmoset monkeys have been proposed as an animal model for coherent visual motion processing[61] and for autism[62]). Finally, the results raise the question of whether perceptual functions can be trained during early phases of brain development (for an analogous example in the attention domain, see ref. [63]).

## Methods

**Participants**. The participants of the EASE sample were predominantly recruited from the greater Stockholm area. Recruitment of EL infants was done via direct outreach and announcements in pediatric clinics, while LL infants were recruited from the Swedish Population Registry (via letters, ~20% response rate). The EL infants were selected for having one or more older siblings diagnosed with ASD. LL infants were screened to include only those who did not have any family member, up to second degree, diagnosed with ASD. More detailed descriptions of the EASE study (e.g., demographics, exclusion criteria, types of tests administered, etc.) can be found in Supplementary Notes 1.

The BATSS cohort comprised same-sex 5-month old infant twin-pairs, screened for general exclusion criteria, such as vision-impairment, premature birth, epilepsy, and medical conditions that might affect brain development, but not for specific neurodevelopmental disorders. Similarly, to the EASE LL group, twin families were recruited from the Swedish Population Registry. Complete demographics and procedures of BATSS study are provided elsewhere[14].

The EASE and BATSS studies were approved by the Stockholm Regional Ethics Committee and all parents of infants signed a written consent for participation in either study. The final tallies of sample size useable for analysis from the two cohorts are: 73 individuals from EASE and 452 individuals from BATSS, see Supplementary Tables 2–4 for detailed breakdown of data attrition (due to exclusions at the analysis level) for each dataset and relevant descriptive statistics.

## Measures

*General EEG setup*. In both cohorts, EEG measurement was performed when infants were 5 months old. In the BATSS cohort, both twins were tested on the same day. The EEG recording was performed using an age-appropriate 128-channel Geodesic Sensor Net HCGSN 130 (Electrical Geodesic Inc., OR; same apparatus and testing room used for both cohorts). Signal sampling rate was 500 Hz using the vertex reference, which was further amplified using EGI Net GES 300 Amp (Electrical Geodesic Inc., OR), and stored for later analysis. Infants were seated on their parents' lap during the experiment.

*Visual stimuli*. For evoking the steady-state visually evoked potentials (SSVEP), the same set of stimuli used in related previous studies[15,16] was employed. The stimulus set has been shown to reliably evoke visual processes responsive to coherent patterns of motion and form[58,59], and similar patterns of topographical EEG activation have been demonstrated in two independent samples of typically-developing infants[15,16]. For evoking form processing (Supplementary Video 1), static short line segments were used and coherent and random patterns (phases) were shown interchanging in succession, with each successive pair of coherent and random phases constituting a cycle of 500-ms (Fig. 1a). In the coherent phase, the line segments were arranged to form concentric circles lasting for 250 ms, while in the random phase the line segments were arranged in random orientations lasting also for 250 ms[15]. For evoking motion processing (Supplementary Video 2), interchanging phases of globally coherent and random dot movements were shown in succession. Each successive pair of coherence and random phases constitute a 500-ms cycle. In the coherent phase, the dots move along a fixed trajectory to form concentric circles lasting for 250 msec, while in the random phase the dots move along random trajectories lasting also for 250 ms. In both motion and form conditions, only the transition from random phase to coherent phase is thought to trigger global visual processing, while each phasic transition (i.e., both random-to-coherent and coherent-to-random) would trigger local visual processing[16,60]. Thus, the stimulus set was designed to evoke SSVEPs both in the frequency of 2 Hz (F1), due to a random-to-coherent transition occurring every 500 ms, indicating global processing activities, and in the frequency of 4 Hz (F2), due to a phasic transition every 250 ms, indicating local processing activities[15,16,60]. Each type of processing (motion or form) was presented separately in ten blocks of 12 s each (a single block consisted of twenty-four 500-ms cycles), giving a total of 240 cycles for each type of visual processing[15]. Thus, the entire stimulus process together produced the four experimental conditions (Global Form, Global Motion, Local Form, Local Motion) that would manifest as 2Hz- and 4Hz-SSVEPs on the EEG[15,16].

*Assessment of autistic traits and symptoms*. In the EASE cohort, the ADOS-2 was administered to infants at 24 and 36 months (by ADOS-2 trainers or ADOS-2 research reliable clinicians) to measure autistic symptoms. In the BATSS cohort, two parent rated questionnaires of infants' social communication skills (the Infant Toddler Checklist, ITC) and infants' autistic traits (the Quantitative Checklist for Autism in Toddlers, Q-CHAT) were administered at 14 months and 24 months, respectively.

**Statistics and reproducibility**. All statistical tests and model-based inferences described here are two-tailed, unless otherwise specified.

*Infant-level and epoch-level exclusions of bad EEG data*. In agreement with previous work[15], we applied a data driven exclusion procedure for EEG data. In brief, because the experiment creates a clear 4-Hz oscillation in midline area of the visual cortex (see Local Form and Local Motion conditions in Fig. 1b and Fig. 2c) whenever the infant is looking at the stimulus, we used the presence of such midline 4-Hz frequency components as an indicator of infants' attention to the stimulus and that he or she was processing the information on the screen. If such a pattern is observable, it is very likely that the infant attended to the stimulus, and because the Global conditions build on the same alternations (using the 2 Hz components), the procedure ensures usable data in all conditions. In the absence of these components, indicating that the infant was not looking at the stimulus or other issues with the recording, we excluded all data from this participant. This exclusion procedure was conducted without any prior knowledge of the results in the 2-Hz conditions or any outcome data. No other criterion was applied to exclude an infant (i.e., if there was a signal of neural visual processing of the stimuli the infant was included, otherwise it was excluded). This procedure resulted in 18 infants being rejected from EASE and 111 infants from BATSS (see Supplementary Table 2). To validate this procedure, we randomly picked 10 participants who were included and 10 who were rejected, 5 from EASE and 5 from BATSS in each case. Using this sample of 20 infants, a simple behavioral coding for infants' looking at the screen during the stimulus presentation was then performed, revealing a statistically significant difference in attention between those included and those rejected, but no attentional difference between EASE and BATSS infants (Supplementary Fig. 3).

Within each infant, epochs having too much contamination by artefacts from blinks and gross movements, both of which produced large and localized deflections in the EEG readings, were excluded during EEG preprocessing: epochs with a voltage range exceeding 100uV were automatically removed. Furthermore, blink artefacts occur very infrequently in infant populations[64] and are also very unlikely to affect our experiment's SSVEP because they need to occur roughly twice or four times per second to do so. Overall, no significant difference was found across groups and across types of visual processing (form, motion) in the average number of epochs contributed per participant, which are detailed in Supplementary Table 5.

*Visualization of overall scalp profiles*. To visualize the overall brain activity, we used the values of T2 circular statistics (T2circ) of the SSVEP[20] from all EEG channels and interpolated gradient values for all scalp locations (Fig. 1b).

*The Laterality Score*. For the main analyses (Fig. 2a, b), we used the activation Laterality Score[15], which was based on T2circ values. As in the previous report[15], nine AOI covering approximately the entire visual cortex in the occipital region of the scalp were used. The Laterality Score was defined as the vector-normalized

difference between the strength of activation (T2circ) in the middle AOI and the maximum activation occurred in the other eight non-central (lateral) AOIs. This measure, thus, represents differential activation of the lateral visual cortex relative to the midline activation.

*Differences between experimental conditions.* Based on the T2circ values across the nine AOIs of the visual cortex, we made a topographical plot for each condition (Fig. 2c). We calculated an L2 (Euclidean) distance from each individual profile-vector to the grand average activation vector (i.e., the nine AOIs are represented in a 9-dimensional vector space) over all conditions, all AOIs, and all participants; the resulting distance scores were then used to compare conditions statistically, first by fitting a linear mixed model to account for the non-independence of observations in BATSS sample and followed by a robust effect-size analysis using the expected marginal means (EMM)[65] method. No *p* value (two-tailed) correction was done since this was a planned comparison[66].

*Association analyses with autistic traits and symptoms.* With EASE data, we regressed behavioral test scores (ADOS-2 CS scores) on Laterality Scores in all four conditions. Analogously, with BATSS data, we regressed ITC Total Scores and QCHAT scores, separately in two models, on Laterality Score in *only* the Global Motion condition. Sex and age at behavioral test (i.e., 36, 14, and 24 months for ADOS-2 CS, ITC, and QCHAT, respectively) were used as covariates. In addition, for the EASE model, group (EL/LL) was included as an extra covariate. In all these models, moderation by sex (and in EASE only, also by group [EL/LL]) was tested. The effects of sex, age at EEG assessment (and in EASE: group [EL/LL]) were regressed out of the Laterality Scores before entering them to the models, to remove any potential confounding by the covariates in the associations. The Ordinary Least Squares (OLS) linear model was used for the EASE data, while linear models with General Estimating Equation (GEE) technique were used for the BATSS data to account for within-family relatedness. As there were more participants participating in the ADOS-2 assessment at 24 months than in the one at 36 months, we additionally tested the association between Global Motion Laterality Scores and ADOS-2 CS at 24 months specifically among the EASE EL group who did not participate at 36 months (Fig. 2a, inset; one-tailed test given directional hypothesis).

As supplementary analyses, we further fitted OLS linear models of the sub-scales of ADOS, Social Affect (SA) and Restricted and Repetitive Behavior, on EASE data; see Supplementary Table 1 for results (not reported in the main text).

*Group comparisons.* We performed a planned comparison of the four groups (the two EASE EL subgroups, EASE LL, and BATSS twins; Fig. 2b) in terms of their Laterality Scores in each condition. Again, a linear mixed model and EMM effect-size analysis were employed to account for the violation of independence assumption (in BATSS sample) and the unbalanced sample sizes. Since the comparison was planned, we performed no *p* value correction[66]. Next, to compare the Normative Sample (see main text) and EL high-ADOS group in terms of the normalized activation strength (T2circ, see above) across the nine AOIs in each condition, the combination of linear mixed model and EMM contrasts were used, followed by a Benjamini–Hochberg *p* value correction to account for the multiple tests (nine in each condition).

*Twin modeling of Laterality Scores.* We used a univariate twin model (ACE/ADE model[67]) to estimate the relative contributions of genetic and environmental influences to variation in the Laterality Scores. The sources of variation in a trait can be divided into genetic influences (A and D; additive and dominance genetic effects, respectively, together constituting the "broad-sense heritability"), shared environment (C; family environment), and unique environment (E; environmental influences that makes twins different from each other, including measurement error). The D and C parameters confound each other in the classical twin design, hence cannot be estimated simultaneously from twin data alone. Since monozygotic (MZ) twins share 100% of their segregating DNA, while dizygotic (DZ) twins on average share 50% of their segregating DNA, a higher within pair similarity among MZ twins than DZ twins suggests genetic contribution to variation in a trait. We regressed out sex and age of first lab visit from the Laterality Scores of each condition and fitted the model on the residuals. With these residualized Laterality Scores, the normality assumption was satisfied for both the Global Form and the Global Motion conditions (Supplementary Fig. 4a, b). In the two local change (Local Form and Motion) conditions, which were slightly skewed, we also applied a square-root transformation that corrected both variables to approximately normal (Supplementary Fig. 4c, d). Model selection was then performed based on a series of likelihood ratio tests or the Akaike Information Criterion/AIC).

**Reporting summary.** Further information on research design is available in the Nature Portfolio Reporting Summary linked to this article.

## Data availability
All data are available from the corresponding author upon reasonable request.

## Code availability
All EEG and statistical analyses in this study utilized standard scripts in MATLAB (https://www.mathworks.com) and R (https://www.r-project.org). All codes are available upon request.

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

## Acknowledgements

The authors would like to thank the participating children and families and the colleagues in EASE and BATSS. This study was funded by Riksbankens Jubileumsfond in collaboration with the Swedish Collegium for Advanced Study (Pro Futura), and the Knut and Alice Wallenberg Foundation; and the Innovative Medicines Initiative 2 Joint Undertaking under grant agreement No 777394. This Joint Undertaking receives support from the European Union's Horizon 2020 research and innovation program and EFPIA and AUTISM SPEAKS, Autistica, SFARI. The work leading to these results was also supported by funds from the European Commission (H2020 project CANDY; Grant No. 847818); and from Marie S. Curie Actions SAPIENS Project (Grant MSCA-ITN-2018 No. 814302). The funders had no role in the design of the study; in the collection, analyses, or interpretation of data; in the writing of the manuscript; or in the decision to publish the results.

## Author contributions

I.H., T.F.Y., and P.N. conceived the research. T.F.Y. oversaw data collection. S.B. was responsible for clinical evaluations. I.H. performed data analysis with support from P.N., S.B., M.T., A.R., and T.F.Y. I.H. and T.F.Y. drafted the manuscript. All authors (I.H., P.N., S.B., M.T., A.R., T.F.Y.) reviewed the manuscript and approved the final version.

## Funding

## Competing interests

The authors declare no competing interests. S.B. discloses that he has in the last 5 years acted as an author, consultant, or lecturer for Medice and Roche. He receives royalties for textbooks and diagnostic tools from Hogrefe (ADOS-2, ADI-R, SRS-2, SCQ), Kohlhammer, and UTB.
