## [Peer Review File · Communications Biology]

Reviewers' comments:

Reviewer #1 (Remarks to the Author):

This study reports some intriguing findings linking lateralisation in visual cortex activation when processing global motion in five months old infants with development of autistic traits and symptoms in later months. The sample overall (considering EASE and BATSS together is large, although not so much for EASE only especially when split into two groups) and the method followed is standardised and overall seem rigorous and sound (although the use of corrections for multiple comparisons in some instances and not in others is not fully convincing, i.e., why some comparisons were planned - thus not requiring corrections for multiple comparisons- and others were not planned, was not explained).

My major comment and what is not fully convincing concerns the results, in that in the first analysis with a smaller sample size of 73 participants split in elevated likelihood (EL) and low likelihood (LL) to develop autism it seems that a strong positive association between laterality scores and ados-2 scores was found (larger lateralisation at 5 months of age was found to be associated with higher scores of ados-2 later at 24 and 36 months), however, this is shown for all sample including both EL and LL and for EL only (no result for LL only was reported that I could see), this suggests that differences in global motion in infancy may be associated to developing autism no matter what the likelihood to develop autism based on other factors is (e.g., presence of autistic siblings), this would suggest that this lateralisation is not specifically associated to a future diagnosis of autism which seems to be supported by the small dataset of autistic diagnosed participants in the extended data figure 2 in the supplemental material. This figure shows that 2 out of 4 infants with an autistic diagnosis had low lateralisation scores within the range of LL infants thus putting in discussion the validity of differences in global motion processing as an early marker of autism. If what the authors wanted to conclude is that the laterality score for global motion is a better early marker for autism when examining the general population then one would have expected a more convincing result for the BATSS in showing a strong association between lateralisation in visual cortex activation in response to global motion and measures of autism, however despite the large sample size the found association is small for the test at 14 months and is absent for another test at 24 months. This overall reduces the ability to draw clear conclusions about how informative differences in global motion processing really are for autism development.

Some methodological decisions remained also unclear, for example, as the authors mentioned "In contrast, global form processing is hardly detectable at the age of 4 months" so why examining 5-month-old infants if a comparison between global motion and global form is performed? If global form is hardly detectable in these infants the comparison of global motion with this condition is not very informative at this age.

The analysis for BATSS was simplified and only examined the association between Laterality Scores in the Global Motion and ITC or Q-CHAT and the reason given is to "To reduce analytic complexity" however given the large sample size in BATSS it would be important to keep the analysis the same to the one done for EASE so that the reader can compare global motion to global form as well as to local processing as if associations between these other visual processing are found with ITC or Q-CHAT then that would contrast the uniqueness of global motion as possible early marker of autism.

Please add effect sizes to the results, e.g., for comparisons Cohen's d

Minor points

This is minor in terms of theoretical and methodological stance but it is now a major point of discussion in the autistic field, nowadays it is preferred to talk of autism, or autistic person etc...rather than people with autism, ASD etc...see here for example: <https://www.autism.org.uk/what-we->

do/help-and-support/how-to-talk-about-autism, so I would revise the terminology throughout the paper with this in mind, similarly replacing "atypicalities" with "differences" would be best

In figure 2 add a clarification about the blue asterisks in the global motion panel

For the sub-sample in figure 2 when association was performed with ados-2 at 24 months, those infants seem to be all EL? Is that because they were only EL or because the association was only performed for EL group and not LL in that group? This needs clarification

"During global motion processing, brain activity in the latter group was characterized by atypically strong extreme left lateral (AOIs A1 and A2: $M = -.120$, $**p = .009$ and $M = -.095$, $*p = .026$, respectively) activation and atypically weak midline" activation should be moved before the parenthesis and just after lateral

Using CI 95% in figure 2 c to represent the variability in the normative sample could be more appropriate than standard deviation

"Notably, the raw bivariate correlation between Global Motion Laterality Scores and ITC total scores was low and only marginally significant ($p = .067$)" it is more appropriate to rephrase this as "it failed to reach significance"

Refer to the table in the supplemental material for details of how many infants EEG data were excluded when discussing this in the manuscript

In the abstract: "and local motion in the visual cortex within a large sample of 5-month-old infants" report exact number for EASE and BATSS, also "Finally, based on an infant twin sample, we show that individual differences in the balance between extrastriate and striate activation during global motion processing reflect genetic variation in the population." What population? General population? What genetic variation? Otherwise this sentence results vague

Reviewer #2 (Remarks to the Author):

This manuscript reports a strong study with 5 month old infants, broadly construed, exploring global motion processing as a function of likelihood for autism. The topic is certainly interesting to a broad array of fields and is grounded in theory related to perceptual development. The data are interesting and novel, and the work appears to have been carefully done. The use of a second sample of infants at elevated risk for autism is an excellent control check for the results. Due to the inherent limitations of EEG techniques, it is not possible to know the neural generators of effects, with the manuscript containing much ascribed, brain location specific terminology. The quality of the work and the importance of the findings outweighs the unnecessarily defined neurological substrates putatively associated with the EEG effects.

My key concern with the manuscript is the use of specific neural substrates throughout when this is impossible to determine from EEG. In this regard, striate and extrastriate cortices are irrelevant to the actual results of the study. Terms such as "lateralized across the visual cortex" are incorrect. The topography of an effect does not imply that the generators are directly beneath those electrodes. The authors are seasoned EEG experts and are aware of this. The literature cited throughout as evidence for specific brain regions (e.g., Braddick & Atkinson, 2011) does not ascribe brain regions. The method is entirely reasonable and informative. The laterality score is still a decent approach for helping to define topographical differences between groups although it absolutely does not "represents differential activation of the lateral visual cortex relatively to the midline activation". Rather, it shows electrophysiological topographical differences. The cited method of analysis by Victor and Mast (1991)

does not enable definitive conclusions on subserving brain regions.

Page 1: There is no definition of global form processing.

Page 2: What does "previously validated EEG experiment" mean? Is this a paradigm that has been used extensively and replicated?

Page 7: Some elaboration on the results for the association with autistic traits in the BATSS sample would be useful. If this is only marginally significant then what does this imply for the findings of the paper?

Discussion: The theories advanced to explain the found effects are reminiscent of the subcortical to cortical change in processing seen with faces, as outlined by Johnson, Senju & Tomalski (2015), albeit in a different area of perceptual space. Making some associations with this literature could be beneficial and could suggest that these developmental changes are in play across a range of perceptual mechanisms.

Discussion: The association with processing information in scotopic conditions is interesting although ruling out novelty and difficulty factors in this literature is challenging.

Discussion: The links with optic flow should be made but with caution. Optic flow and motion processing are different aspects of visual processing and should not be conflated.

Methods: The data in Extended Data Table 1 does not convey information on attrition by group for this dataset. It conveys the general information on EASE and BATTSS. It would be of particular importance that a similar amount of attrition is present in each grouping of infants given that the ultimate n in some groups is quite small – such as the LL Controls.

Methods: What were the final contributions by group to the number of analysed cycles for each type of visual processing? The question is raised to explicitly explore the quantity of data per group as an explanation for differences between groups. The authors may wish to consider including some measures of data quality in the manuscript.

Methods: The lack of behavioural coding for visual examination of the stimulus is quite novel. The justification that inclusion was based on the presence of a topographical effect related to visual attention is reasonable but should be justified with some quantitative underpinnings. For example, were infants who did not look producing a different topographical effect? How were behaviours such as eye blinks and other behavioural artefacts accounted for if behaviour was not assessed?

Methods: Did the square root transformation for the two local change conditions correct the skew? This should be explicitly stated.

Dear Editor,

Thank you very much for the opportunity to resubmit our manuscript. Thanks also to the reviewers for very valuable comments, which we have addressed below. We hope the manuscript is now acceptable for publication in Communications Biology.

Best regards

Irzam Hardiansyah,
On behalf of all authors

Reviewers' comments:

Reviewer #1 (Remarks to the Author):

This study reports some intriguing findings linking lateralisation in visual cortex activation when processing global motion in five months old infants with development of autistic traits and symptoms in later months. The sample overall (considering EASE and BATSS together is large, although not so much for EASE only especially when split into two groups) and the method followed is standardised and overall seem rigorous and sound (although the use of corrections for multiple comparisons in some instances and not in others is not fully convincing, i.e., why some comparisons were planned -thus not requiring corrections for multiple comparisons- and others were not planned, was not explained).

#1 - My major comment and what is not fully convincing concerns the results, in that in the first analysis with a smaller sample size of 73 participants split in elevated likelihood (EL) and low likelihood (LL) to develop autism it seems that a strong positive association between laterality scores and ados-2 scores was found (larger lateralisation at 5 months of age was found to be associated with higher scores of ados-2 later at 24 and 36 months), however, this is shown for all sample including both EL and LL and for EL only (no result for LL only was reported that I could see), this suggests that differences in global motion in infancy may be associated to developing autism no matter what the likelihood to develop autism based on other factors is (e.g., presence of autistic siblings), this would suggest that this lateralisation is not specifically associated to a future diagnosis of autism which seems to be supported by the small dataset of autistic diagnosed participants in the extended data figure 2 in the supplemental material. This figure shows that 2 out of 4 infants with an autistic diagnosis had low lateralisation scores within the range of LL infants thus putting in discussion the validity of differences in global motion processing as an early marker of autism. If what the authors wanted to conclude is that the laterality score for global motion is a better early marker for autism when examining the general population then one would have expected a more convincing result for the BATSS in showing a strong association between lateralisation in visual cortex activation in response to global motion and measures of autism, however despite the large sample size the found association is small for the test at 14 months and is absent for another test at 24 months. This overall reduces the ability to draw clear conclusions about how informative differences in global motion processing really are for autism development

Reply to Reviewer: Thank you for these comments! We think 2 issues are important to clarify in response to this:

1) we are not claiming that what we have found is an early marker of later autism diagnosis, because we do not have a sufficient number of cases to evaluate this question. However, some models of autism see 'clinical autism' (ASD) as the extreme end of a continuum of traits spanning the whole population (e.g., (Robinson et al., 2011)), Hence, it is possible to study dimensional autistic traits, and not only categorical autism, i.e., ASD. We study autistic traits dimensionally, and in the EASE sample we map the infant marker to a gold standard measure of autism symptoms – the ADOS (a structured play observation conducted with each child by experienced clinical psychologists who were trained for research reliability on the instrument). We want to highlight that even within the group of EL infants (alone), early global motion processing explained around 20% of variance in later ADOS scores. While, as for all findings, this should be replicated in independent samples, the strength of this relationship between a specific infant phenotype and later autism (ADOS) is - to our knowledge - unprecedented in the field. In the context of the theoretical background - which hypothesizes a link between this phenotype and autism – we argue that this result alone is a strong indication that atypicalities in global motion processing in infancy is linked to later autism symptoms (which, again, is not the same as diagnosis, but correlated with it).

2) It is expected that in a sample of elevated likelihood for ASD, there will be many cases with elevated traits but who do not meet the categorical cut-off for ASD. Hence, it is not surprising that one sees dimensional associations in the EL sample that are not seen to the same extent in the LL group. We think that our results are fully in line with the view that global motion processing is linked to autism dimensionally. In BATSS, we used a parent questionnaire which, while tapping into similar constructs (autistic traits), cannot be equated in terms of either reliability or clinical validity (see e.g., (Allison et al., 2021)). Thus, while in principle we agree with the reviewer that a similar association should be present in all samples, there could be methodological reasons why we do not see it in the BATSS sample. Further, a dimensional view of autism does not necessarily entail that a similar, linear association should be observed across all of the population, and independently of genetic liability. Thus, the fact that we do not observe similar strong associations in BATSS is informative, as it suggests that it is in EL groups the association is observable. This even has clinical implications, as it suggests that the marker would work best as a predictor of later development if combined with other information, such as family history of ASD. We also discuss the meaning of this negative finding in terms of the continuity model of autism in the Discussion section, p.13.

#2 - Some methodological decisions remained also unclear, for example, as the authors mentioned "In contrast, global form processing is hardly detectable at the age of 4 months" so why examining 5-month-old infants if a comparison between global motion and global form is performed? If global form is hardly detectable in these infants the comparison of global motion with this condition is not very informative at this age.

Reply to Reviewer: This is a good point. However, while it is possible that global form is simply not mature enough (in any group) to reveal group differences, previous literature (e.g., (O. Braddick et al., 2006; Palomares et al., 2010)) suggest that GF elicits distinct

neural signatures at 5 months. And indeed, we show in our data that GF is distinct from both local form and Global motion processing. We have added a sentence on this point in the Discussion section, p.10.

#3 - The analysis for BATSS was simplified and only examined the association between Laterality Scores in the Global Motion and ITC or Q-CHAT and the reason given is to “To reduce analytic complexity” however given the large sample size in BATSS it would be important to keep the analysis the same to the one done for EASE so that the reader can compare global motion to global form as well as to local processing as if associations between these other visual processing are found with ITC or Q-CHAT then that would contrast the uniqueness of global motion as possible early marker of autism.

Reply to Reviewer: Thank you for this point. We have now added these analyses. They revealed that in the BATSS sample, the overall explanatory power of the EEG measures is in fact negligible, and we have revised the Results section, p.9, of the manuscript accordingly.

#4 - Please add effect sizes to the results, e.g., for comparisons Cohen’s d

Reply to Reviewer: We have added effect sizes for all reported mean differences (with Cohen’s d) and regression coefficients (with partial η^2), but not for heritability and other twin modelling parameters because it is not usually done for these parameters. Pearson’s (bivariate) correlations are an effect size, by definition. These effect sizes are reported in the Results section, Fig. 2, pp.8-9, of the main text, as well as in Supplementary Table 1, next to the corresponding results.

Minor points

#5 - This is minor in terms of theoretical and methodological stance but it is now a major point of discussion in the autistic field, nowadays it is preferred to talk of autism, or autistic person etc...rather than people with autism, ASD etc...see here for example: <https://www.autism.org.uk/what-we-do/help-and-support/how-to-talk-about-autism>, so I would revise the terminology throughout the paper with this in mind, similarly replacing “atypicalities” with “differences” would be best

Reply to Reviewer: Thank you for this comment, we have tried our best to modify the language throughout.

#6 - In figure 2 add a clarification about the blue asterisks in the global motion panel

Reply to Reviewer: We added the information at the end of the caption of Fig. 2 (pp.8-9).

#7 - For the sub-sample in figure 2 when association was performed with ados-2 at 24 months, those infants seem to be all EL? Is that because they were only EL or because the association was only performed for EL group and not LL in that group? This needs clarification

Reply to Reviewer: Yes, for this analysis (which was restricted to the ones not available for 3 year follow up), we did the analysis only within the EL group. The reasons for this was i) in general, the EL includes much more variable developmental trajectories, and hence more interesting and feasible to study dimensional in small samples ii) there were only 3 infants in the LL group who did only participate at 24 month, hence we did not consider this a meaningful sub sample to include in this analysis. We acknowledge that this analysis is based on a very small sample, but in light of the main result/scatterplot in panel (a) of the figure, it still contributes to strengthening the conclusion that GMP is related to later autism symptoms in the EL group. See also our response to question #1 above.

*#8 - "During global motion processing, brain activity in the latter group was characterized by atypically strong extreme left lateral (AOIs A1 and A2: $M = -.120$, $**p = .009$ and $M = -.095$, $*p = .026$, respectively) activation and atypically weak midline" activation should be moved before the parenthesis and just after lateral*

Reply to Reviewer: Thank you for pointing this out. The referenced part of Fig. 2 caption (pp.8-9) has been updated accordingly.

#9 - Using CI 95% in figure 2 c to represent the variability in the normative sample could be more appropriate than standard deviation

Reply to Reviewer: We have updated Fig. 2c and its caption (pp.8-9) in the manuscript accordingly.

#10 - "Notably, the raw bivariate correlation between Global Motion Laterality Scores and ITC total scores was low and only marginally significant ($p = .067$)" it is more appropriate to rephrase this as "it failed to reach significance"

Reply to Reviewer: In light of the updated results for this part, on p.9 of manuscript, we have removed the sentence entirely.

#11 - Refer to the table in the supplemental material for details of how many infants EEG data were excluded when discussing this in the manuscript

Reply to Reviewer: Thank you for pointing this out. Whenever we mentioned exclusions in the main text, we made an explicit reference to the Extended Data Table 1 (as well as Extended Data Tables 2 & 3 where applicable).

#12 - In the abstract: "and local motion in the visual cortex within a large sample of 5-month-old infants" report exact number for EASE and BATSS, also "Finally, based on an infant twin sample, we show that individual differences in the balance between extrastriate and striate activation during global motion processing reflect genetic variation in the population." What population? General population? What genetic variation? Otherwise, this sentence results vague

Reply to Reviewer: Thank you for these good points. We have added the requested numbers in and have chosen to delete the referred sentence from the Abstract.

Reviewer #2 (Remarks to the Author):

This manuscript reports a strong study with 5 month old infants, broadly construed, exploring global motion processing as a function of likelihood for autism. The topic is certainly interesting to a broad array of fields and is grounded in theory related to perceptual development. The data are interesting and novel, and the work appears to have been carefully done. The use of a second sample of infants at elevated risk for autism is an excellent control check for the results. Due to the inherent limitations of EEG techniques, it is not possible to know the neural generators of effects, with the manuscript containing much ascribed, brain location specific terminology. The quality of the work and the importance of the findings outweighs the unnecessarily defined neurological substrates putatively associated with the EEG effects.

#1 - My key concern with the manuscript is the use of specific neural substrates throughout when this is impossible to determine from EEG. In this regard, striate and extrastriate cortices are irrelevant to the actual results of the study. Terms such as “lateralized across the visual cortex” are incorrect. The topography of an effect does not imply that the generators are directly beneath those electrodes. The authors are seasoned EEG experts and are aware of this. The literature cited throughout as evidence for specific brain regions (e.g., Braddick & Atkinson, 2011) does not ascribe brain regions. The method is entirely reasonable and informative. The laterality score is still a decent approach for helping to define topographical differences between groups although it absolutely does not “represents differential activation of the lateral visual cortex relatively to the midline activation”. Rather, it shows electrophysiological topographical differences. The cited method of analysis by Victor and Mast (1991) does not enable definitive conclusions on subserving brain regions.

Reply to Reviewer: While we agree in principle, we want to stress that the particular EEG experiments with global motion and form has been used extensively before (by employing forced choice motion discrimination paradigm (Britten et al., 1992), e.g., random dot kinematogram (RDK), from which the paradigm used in our study was derived; the paradigm reported here, as well as in (Wattam-Bell et al., 2010) and (Nyström et al., 2021) is an ‘infant-friendly’ version of this classical paradigm, which allowed it to be used in passive-viewing tasks; see also reply to Comment #3 below), and that rather detailed links between activation patterns and underlying sources have been made, including in studies with fMRI and invasive techniques (e.g., (Baker et al., 1991; Born & Bradley, 2005; Braddick et al., 2000; Braddick & Qian, 2001; Livingstone & Hubel, 1987; Robertson et al., 2014); see also (Braddick & Atkinson, 2007) for an extensive review in developmental context). For the case of our paradigm, Wattam-Bell and colleagues (Wattam-Bell et al., 2010) in their Supplementary Information (section “From Scalp Topographies to Cortical Sources”) made a long argument for why central versus lateralized activation in this paradigm reflects differences in activation in the sources rather than mere artifactual EEG patterns. Based on such extensive evidence and arguments already present in the current literature, we feel that our discussion around possible brain sources of the observed activation patterns is appropriate.

#2 - Page 1: There is no definition of global form processing.

Reply to Reviewer: We have added this in the Introduction section, p.3, of the manuscript.

#3 - Page 2: What does “previously validated EEG experiment” mean? Is this a paradigm that has been used extensively and replicated?

Reply to Reviewer: Thank you for this question, indeed it is what we meant. The same paradigm has been used in a number of previously published journal articles to study the same visual perceptual processes (global/local form & motion processing) in infants (e.g., (O. Braddick et al., 2006; O. J. Braddick et al., 2006; Nyström et al., 2021; Wattam-Bell et al., 2010)).

In particular, as we also mention in one passage of the Discussion section (p.13), the transitions from random to coherent phase (but not on the opposite direction, i.e. from coherent to random) of both motion and form stimuli in this paradigm reliably evokes SSVEPs with a frequency reflecting these transitions (here at 2 Hz) (Braddick & Atkinson, 2007), whose power amplitude has been shown to correlate with the amount of coherence in the classical threshold-based (e.g., random dot kinematogram / RDK) paradigm - and more importantly, to be absent when the amount of coherence is 0% (movement directions / segment orientations are purely random) (Braddick & Atkinson, 2007; O. Braddick et al., 2006; O. J. Braddick et al., 2006). Therefore, since the amount of coherence during the coherent phase is constant and 100% (i.e., no noise), the presence of such frequency component (at 2 Hz) in the induced SSVEP is taken as an indicator that the global visual processing occurs (see also (Wattam-Bell et al., 2010): Supplementary Information section “F1 as a Signature of Global Processing”; and (Braddick & Atkinson, 2007): pp.157-160).

As for the replicability, using the same paradigm as ours, previous studies have reported similar patterns of activation in both the global motion and global form conditions with at least 2 independent samples of typically-developing infants, i.e., (Wattam-Bell et al., 2010), Fig. 3, p.413 and (Nyström et al., 2021), Fig. 2, p.4.

We have added the above information to explain about the validity of our paradigm in the Discussion section, p.10, and in the Method section, under “Visual stimuli”, p.16.

#4 - Page 7: Some elaboration on the results for the association with autistic traits in the BATSS sample would be useful. If this is only marginally significant then what does this imply for the findings of the paper?

Reply to Reviewer: In response to Reviewer 1 we have expanded and repeated the analysis of the link between EEG activity and later traits. Considering that the results of this analysis shows that very little of the variance in later parent-rated traits can be explained by the EEG data, our conclusion regarding this part has changed as well. We offer 2 explanations, that are not mutually exclusive a) parent ratings of autistic traits (BATSS) are less reliable than clinician rated symptoms based on direct observation (EASE), and b) it is possible that the association between GMP and later autism is much more pronounced in infants at elevated likelihood, due to their increased risk and more variable neurodevelopmental trajectories. We have also now added a new reference, i.e., (Allison et al., 2021), to support our argument in (a).

#5 - Discussion: The theories advanced to explain the found effects are reminiscent of the subcortical to cortical change in processing seen with faces, as outlined by Johnson, Senju & Tomalski (2015), albeit in a different area of perceptual space. Making some associations with this literature could be beneficial and could suggest that these developmental changes are in play across a range of perceptual mechanisms.

Reply to Reviewer: Thank you, we have added this reference together with a previous one on this topic in the Discussion (p.11, third paragraph).

#6 - Discussion: The association with processing information in scotopic conditions is interesting although ruling out novelty and difficulty factors in this literature is challenging.

Reply to Reviewer: We agree. Still, it is worth noting this striking similarity so that readers can more easily make informed inferences about possible explanations for our result. We have made some additions to this interpretation to make it more nuanced.

#7 - Discussion: The links with optic flow should be made but with caution. Optic flow and motion processing are different aspects of visual processing and should not be conflated.

Reply to Reviewer: Thank you for this very good note. To avoid confusion on these different processes, we have decided to remove the phrase entirely.

#8 - Methods: The data in Extended Data Table 1 does not convey information on attrition by group for this dataset. It conveys the general information on EASE and BATTs. It would be of particular importance that a similar amount of attrition is present in each grouping of infants given that the ultimate n in some groups is quite small – such as the LL Controls.

Reply to Reviewer: Thank you for pointing this out. This information is available in Extended Data Tables 2 and 3, showing how many participants remained (equivalently, dropped out) for each group during different stages of EEG & behavioural assessments.

#9 - Methods: What were the final contributions by group to the number of analysed cycles for each type of visual processing? The question is raised to explicitly explore the quantity of data per group as an explanation for differences between groups. The authors may wish to consider including some measures of data quality in the manuscript.

Reply to Reviewer: Thank you very much for this good point. The average number of analysed cycles contributed by each participant does not significantly differ either across groups (LL controls, EL Lo-ADOS, EL Hi-ADOS, Twins) or across types of visual processing (Form & Motion), as supported by the result of a 2-way ANOVA analysis of the number of VEP cycles obtained from the participants (see figure below). On average, individuals in all groups contributed around 150-170 cycles out of a total 200 cycles (15-25% cycle exclusion due to artifact rejection) in each group and each processing type. We have also added this information in the Method section, pp.18-19, of the manuscript as well as in Supplementary Table 2.

#10 - *Methods: The lack of behavioural coding for visual examination of the stimulus is quite novel. The justification that inclusion was based on the presence of a topographical effect related to visual attention is reasonable but should be justified with some quantitative underpinnings. For example, were infants who did not look producing a different topographical effect? How were behaviours such as eye blinks and other behavioural artefacts accounted for if behaviour was not assessed?*

Reply to Reviewer: Thank you for pointing this out, this is an important issue, and we missed to include this information while restructuring the article from an earlier draft. As in previous studies (Nyström et al, 2021), we used an automatic exclusion for epochs with voltage ranges >100uV, and the remaining epoch numbers for the different groups and conditions can be found in Supplementary Table 2. As for eye blinks, they occur very infrequently in the infant population, and at 5 months the spontaneous blink rate is approximately 2 blinks per minute (Cruz et al., 2011). Even if blinks did pass the 100uV threshold for a very small number of epochs, their impact would be minimal for the T2Circ statistics at 2Hz or 4Hz. The same is also true with gross movements. We have added this information in the Method section, p.18, of the manuscript.

Importantly, in response to your comment, a simple behavioural attention coding on a small sample of 20 infants who were included (10) in and rejected/excluded (10) from the analysis. Indeed, we found a statistically significant difference in attention towards the screen (i.e., looking at the stimulus) between the rejected infants (due to having a flat topographical profile) and those included (due to having strong middle activation for local (4Hz) form or motion condition; see panel (a) of the plot below). Importantly, we did not find any significant difference in this behavioural looking between EASE and BATSS infants; see panel (b) of the plot. *Note:* For both panels, scale of attention level: 0 = lowest (not looking at all) - 3 = highest (always / almost always looking). We have added the plot as Extended Data Fig. 5 in the manuscript.

#11 - Methods: Did the square root transformation for the two local change conditions correct the skew? This should be explicitly stated.

Reply to Reviewer: Thank you for pointing this out. We have added such a statement in the Method section, p.21, of the manuscript.

References

- Allison, C., Matthews, F. E., Ruta, L., Pasco, G., Soufer, R., Brayne, C., Charman, T., & Baron-Cohen, S. (2021). Quantitative Checklist for Autism in Toddlers (Q-CHAT). A population screening study with follow-up: the case for multiple time-point screening for autism. *BMJ paediatrics open*, 5(1).
- Baker, C., Hess, R. F., & Zihl, J. (1991). Residual motion perception in a "motion-blind" patient, assessed with limited-lifetime random dot stimuli. *Journal of Neuroscience*, 11(2), 454-461.
- Born, R. T., & Bradley, D. C. (2005). Structure and function of visual area MT. *Annu. Rev. Neurosci.*, 28, 157-189.
- Braddick, O., & Atkinson, J. (2007). Development of brain mechanisms for visual global processing and object segmentation. *Progress in Brain Research*, 164, 151-168.
- Braddick, O., Birtles, D., Warshafsky, J., Akthar, F., Wattam-Bell, J., & Atkinson, J. (2006). Evoked potentials specific to global visual coherence in adults and infants. *Perception ECVF abstract*, 35, 0-0.
- Braddick, O., O'Brien, J., Wattam-Bell, J., Atkinson, J., & Turner, R. (2000). Form and motion coherence activate independent, but not dorsal/ventral segregated, networks in the human brain. *Current Biology*, 10(12), 731-734.
- Braddick, O., & Qian, N. (2001). The organization of global motion and transparency. In *Motion vision* (pp. 85-112). Springer.
- Braddick, O. J., Birtles, D., Mills, S., Warshafsky, J., Wattam-Bell, J., & Atkinson, J. (2006). Brain responses to global perceptual coherence. *Journal of Vision*, 6(6), 426-426.
- Britten, K. H., Shadlen, M. N., Newsome, W. T., & Movshon, J. A. (1992). The analysis of visual motion: a comparison of neuronal and psychophysical performance. *Journal of Neuroscience*, 12(12), 4745-4765.

- Cruz, A. A., Garcia, D. M., Pinto, C. T., & Cechetti, S. P. (2011). Spontaneous eyeblink activity. *The ocular surface*, 9(1), 29-41.
- Livingstone, M. S., & Hubel, D. H. (1987). Psychophysical evidence for separate channels for the perception of form, color, movement, and depth. *Journal of Neuroscience*, 7(11), 3416-3468.
- Nyström, P., Jones, E., Darki, F., Bölte, S., & Falck-Ytter, T. (2021). Atypical topographical organization of global form and motion processing in 5-month-old infants at risk for Autism. *Journal of autism and developmental disorders*, 51, 364-370.
- Palomares, M., Pettet, M., Vildavski, V., Hou, C., & Norcia, A. (2010). Connecting the dots: How local structure affects global integration in infants. *Journal of Cognitive Neuroscience*, 22(7), 1557-1569.
- Robertson, C. E., Thomas, C., Kravitz, D. J., Wallace, G. L., Baron-Cohen, S., Martin, A., & Baker, C. I. (2014). Global motion perception deficits in autism are reflected as early as primary visual cortex. *Brain*, 137(9), 2588-2599.
- Robinson, E. B., Koenen, K. C., McCormick, M. C., Munir, K., Hallett, V., Happé, F., Plomin, R., & Ronald, A. (2011). Evidence that autistic traits show the same etiology in the general population and at the quantitative extremes (5%, 2.5%, and 1%). *Archives of general psychiatry*, 68(11), 1113-1121.
- Wattam-Bell, J., Birtles, D., Nyström, P., Von Hofsten, C., Rosander, K., Anker, S., Atkinson, J., & Braddick, O. (2010). Reorganization of global form and motion processing during human visual development. *Current Biology*, 20(5), 411-415.

Reviewers' comments:

Reviewer #1 (Remarks to the Author):

The authors have addressed all my points and thank them for the in-depth responses and consideration and interesting discussion. I have no further comment.

Reviewer #2 (Remarks to the Author):

The authors have done a good job of revising the manuscript based on earlier comments.

My prior concern related to putative neural generators still stands. A discussion around possible sources of activity is reasonable but the manuscript is too definitive in this regard - (e.g., "it is source X" when it can only be concluded "it may be source X"). It is of note that the cited articles about source are not developmental in nature. Despite this disagreement that I have with the authors, the paper itself is sound and of strong interest.

The further information on the paradigm and prior replication is very helpful.

The new analysis between EEG results and later traits is quite different to the earlier version. The information is nonetheless still new and interesting for the field in my view in addition to generating more questions that need answering.

The analysis related to data quality and attrition rates strengthens the paper significantly.

Dear Reviewers,

Thank you very much for your further comments and feedback. Please find below our responses to them.

Sincerely Yours,

Irzam Hardiansyah,
On behalf of all authors

Reviewers' comments:

Reviewer #1 (Remarks to the Author):

The authors have addressed all my points and thank them for the in-depth responses and consideration and interesting discussion. I have no further comment.

Reply to Reviewer: We sincerely thank you very much for all your comments and very constructive feedback which allowed us to refine this manuscript into the current final form.

Reviewer #2 (Remarks to the Author):

The authors have done a good job of revising the manuscript based on earlier comments.

My prior concern related to putative neural generators still stands. A discussion around possible sources of activity is reasonable but the manuscript is too definitive in this regard - (e.g., "it is source X" when it can only be concluded "it may be source X"). It is of note that the cited articles about source are not developmental in nature. Despite this disagreement that I have with the authors, the paper itself is sound and of strong interest.

The further information on the paradigm and prior replication is very helpful.

The new analysis between EEG results and later traits is quite different to the earlier version. The information is nonetheless still new and interesting for the field in my view in addition to generating more questions that need answering.

The analysis related to data quality and attrition rates strengthens the paper significantly.

Reply to Reviewer: We understand and accept the Reviewer's concern regarding the neural sources of our visual cortical EEG laterality patterns. Therefore, we have did our best to change the language that we used to discuss this issue to emphasize it's probabilistic/uncertain nature. Hopefully, these changes (can be found throughout the Results and Discussion sections of the manuscript, but mostly in Discussion section) would be deemed sufficient to assuage the Reviewer's concern. We will be more careful with this issue in the future.

We also sincerely thank you very much for all your comments and very constructive feedback which allowed us to refine this manuscript into the current final form.